# Study on the Mechanism of Rainfall-Runoff Induced Nitrogen and Phosphorus Loss in Hilly Slopes of Black Soil Area, China

Tienan Li [1], Fang Ma [2,*], Jun Wang [1], Pengpeng Qiu [1], Ning Zhang [1], Weiwei Guo [1], Jinzhong Xu [1] and Taoyan Dai [3]

1   Heilongjiang Province Hydraulic Research Institute, Harbin 150080, China
2   State Key Laboratory of Urban Water Resources and Environment, Harbin Institute of Technology, Harbin 150090, China
3   College of Water Conservancy and Hydropower, Heilongjiang University, Harbin 150080, China
*   Correspondence: mafang@hit.edu.cn; Tel.: +86-451-86283787

**Abstract:** In order to identify the effects of the slope and precipitation intensity on the soil runoff depth and runoff rate, different tillage patterns (slope-ridge direction, horizontal slope-ridge direction, no-ridge farming) and different slopes (3° and 5°) were set up, and five typical rainfalls from June to September 2021 were selected, to dynamically monitor the soil-erosion dynamics of the test plots under different rainfall intensities. The results show that cross-slope-ridge cropping has a retention effect on runoff, which effectively inhibits the ineffective loss of rainfall confluence. Among these results, the variation range in the soil runoff depth under cross-slope-ridge treatment conditions was 0.11~0.94 mm, while that under the slope-ridge treatment and no-ridge treatment conditions was increased to 1.44~12.49 mm and 3.45~14.96 mm, respectively. It found that the loss of soil nutrients was significantly higher in the slope-ridge direction and in the no-ridge farming condition than in the horizontal slope-ridge direction. It is worth noting that, as the slope of the cultivated land increases, the erosive capacity of the precipitation runoff for the soil phosphorus increases, while the carrying capacity of the soil nitrogen decreases, and the correlation analysis results confirm that the corresponding relationship between the free diffusion capacity of the soil ammonium nitrogen and soil erosion is weaker than that between the nitrate nitrogen and soil erosion. The effects of single factors, such as the slope, ridge direction, and precipitation intensity of the cultivated land, have a significant impact on the soil water- and fertilizer-loss process, while the influence effect of the multi-factor coupling process on soil erosion is weakened. It was confirmed that the erosion process of rainfall runoff on soil nitrogen and phosphorus loss in slope cultivated land is the result of multi-factor action, and the artificial modification of the tillage mode can effectively regulate the effect of farmland water and fertilizer loss.

**Keywords:** slope cultivated land; farming patterns; surface runoff; loss of nitrogen and phosphorus

## 1. Introduction

Black soil, deemed as a "giant panda in the arable land", is a precious land resource on earth, with a black or dark-brown humus surface layer, which is rich in nutrients, and has high fertility levels and a wide seedability, especially for soybeans, corn, cereals, wheat, and other growth [1,2]. Thus, the accelerated exploitation of black soil land resources has gradually increased, due to the increasing demand for food, particular in northeast China [3]. The excessive mining scale and unreasonable irrigation methods used during agricultural production produce residual chemicals in the soil, which can be washed into water bodies by rainfall, leading to water environmental pollution. Meanwhile, the organic matter content of black soil land decreases as the rain washes away, which can lead to the nutrients becoming infertile gradually, resulting in a serious threat to the security of food production [4]. Moreover, the wanton exploitation of agricultural resources by human

beings has accelerated soil erosion, and promoted the development of erosion ditches, resulting in an enhancement in the washing and transportation capacity of atmospheric precipitation toward the soil, while soil solutes converge into water bodies with the effects of soil erosion, resulting in frequent environmental pollution in rivers and lakes [5,6].

Nitrogen and phosphorus in soil are indispensable nutrients in the good growth of crops, and an appropriate amount of soil fertilizer supplementation can effectively promote the growth and development of crops [7]. In order to increase crop yields, soil fertilizers are often applied excessively, while precipitation generates runoff that causes soil nutrient loss, and wash excess fertilizers from the soil into low-lying areas, and eventually into water bodies, leading to the eutrophication of water bodies, and the degradation of the water environment [8,9]. The nitrogen and phosphorus export processes on small watersheds vary significantly, depending on cropping patterns. The intensity of the nitrogen and phosphorus export increases with the rainfall intensity, and the occurrence of stormwater runoff increases the risk of downstream eutrophication. It was found that rainfall, extreme rainfall events, the soil type, subsurface characteristics, and human activities affect the surface runoff nitrogen and phosphorus loss from watersheds [10,11]. Therefore, the prevention, control, and treatment of agricultural nonpoint source pollution are still important issues restricting the healthy and sustainable development of agriculture [12,13].

However, the pollution of agricultural surface sources is a complex mechanism, which requires an effective method of identifying the effects of rainfall and climate change on soil. In general, in the northern seasonal permafrost area, soil erosion is the main control factor for agricultural nonpoint source pollution [14]. In addition, the soil erosion types and complex processes in black soil areas are manifested through the characteristics of multiforce coupling and multi-process superposition [15]. Affected by the seasonal freeze–thaw cycle, the soil porosity increases, large pores and fissures continue to develop, the stability of the soil aggregates decreases, the soil erosion resistance weakens, and the soil hydraulic erosion intensifies [16]. With the increase in the water flow intensity, fine trenches gradually develop in the soil, and the eroded sediment between the fine trenches is continuously transported to wide trenches, and the effect of the cultivated land soil erosion is significantly increased [17,18]. In addition, soil erosion is complicated by short-term rainfall in summer, and the soil structure is changed by short-term heavy rainfall, which promotes changes in the surface and underground sand production characteristics of sloped cultivated land, and the modification force becomes the main driving force behind surface erosion [19].

In recent years, scholars have carried out a large amount of exploration and research into the control factors behind soil erosion, and have confirmed that human transformation activities, such as farmland soil tillage measures, straw return, crop rotation systems, and forest belt construction, have had a positive impact on the prevention and control of soil erosion [20–22]. Cultivation practices significantly change the soil's physicochemical properties, and regulate slope hydrological processes; as proposed by Ahmed et al. [23], transverse ridge cropping significantly increases the surface roughness, increases the capacity for surface runoff accumulation, and improves the soil and water conservation capacity of sloped cultivated land. After the decomposition of stubble and straw, the soil organic matter content increased, the cohesion ability of small-particle soil aggregates was improved, the soil stability was enhanced, and the soil erosion resistance effect was correspondingly improved [24]. Forest belt construction can block and absorb surface runoff, conserve water sources, and inhibit the germination and development of fine soil trenches in farmland [25]. In addition, the density of fine trenches decreases with the increase in the forest belt density, but the influence of the forest belt on fine furrow erosion shows a stable–decreasing–disappearing trend with the increase in the forest belt distance [26]. Based on the above analysis, it can be seen that previous scholars have carried out a lot of research on the soil erosion process and mechanism, the main environmental factors controlling soil erosion, and the coupling effect of the soil erosion environment, but there are few studies on the synergistic response relationship between nitrogen and phosphorus loss and rainfall runoff that is involved in the soil erosion process.

This study uses a one-way analysis of variance method to address the study of the migration pattern of soil nitrogen and phosphorus in hilly and diffuse arable land in the northeastern black soil region during rainfall runoff, which can respond to the impact of human activities on the water environment and, thus, contribute to the improvement of the source control of agricultural surface source pollution. It aims to effectively prevent and control the effects of water and soil environmental pollution caused by human agricultural activities, and provide data-based support for the healthy and sustainable use of cultivated land on black soil slopes.

## 2. Materials and Methods

### 2.1. Overview of the Study Area

The experimental area is locating in the runoff field of the Heilongjiang Institute of Soil and Water Conservation, located on the south bank of the Songhua River, belonging to the southern part of the Songnen Plain, with the geographical coordinates E 127°24′47″, N 45°44′57″. The landform belongs to the hilly type, the terrain slopes from north to south, and the overall trend from northwest high to southeast low shows a changing trend; the regional location map is detailed in Figure 1. The study area is located in the middle temperate continental monsoon climate zone, with high temperatures, a rainy summer, and a cold and dry winter. The average minimum temperature in winter is −37.6 °C, the average maximum temperature in summer is 37.2 °C, and the average annual temperature is 4.1 °C. The average annual precipitation is 681 mm, the precipitation is mainly concentrated in June~August, and the annual runoff depth is 90 mm. After manual sampling and analysis, it can be found that the 0~35 cm soil layer in this area is black loam, and the soil layer below 35 cm is dark-brown clay. The soil pH value was 6.65, the organic matter content was 41.32 g/kg, and the soil total nitrogen and total phosphorus contents were 2.03 and 1.59 g/kg, respectively. In addition, the soil alkaline hydrolysis nitrogen content was 172.41 mg/kg, available phosphorus 41.12 mg/kg, and available potassium 213.14 mg/kg. The soil texture of the test area is fertile and suitable for crop growth. However, with the increase in the intensity of agricultural water and soil resource development, the regional surface vegetation and underlying surface have been seriously damaged, and the area has become a key area for soil erosion control in the national black soil area. In order to respond to the erosion problem in the region, the one-way analysis of variance method is introduced, to analyze the migration pattern of soil nitrogen and phosphorus in hilly diffuse arable land during rainfall runoff.

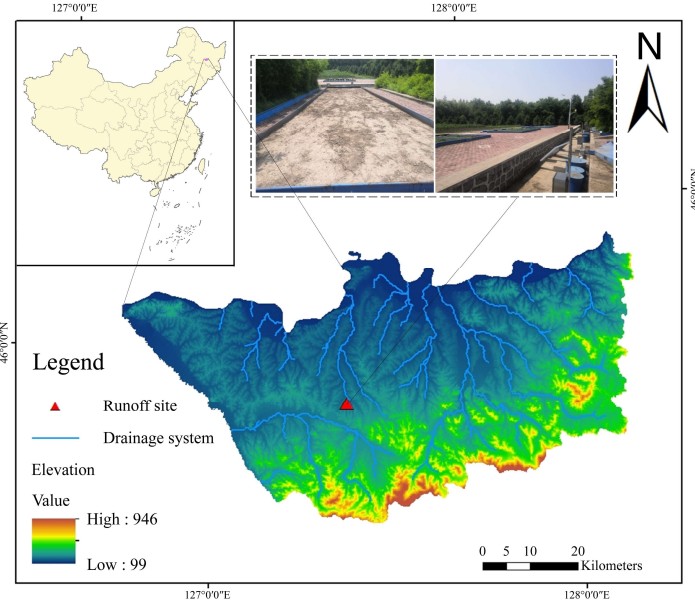

**Figure 1.** Location of the study area.

### 2.2. Runoff Site Layout

Considering the topographic characteristics of sloping land in the black soil area, and the characteristics of the significance of rainfall production flow differences, the slope of the experimental community was set to 3° and 5°, respectively, and each slope cell was set with the three methods of cross-ridge, along-the-ridge, and no-ridge, resulting in a total of six treatments (Table 1). The dimensions of each runoff cell were set to 5 × 20 m, the ridge width and height of the cross and trail ridges were set to 60 cm, and the height of the ridge was 15 cm (Figure 2). According to the local traditional farming method, the plot was uniformly rotated before sowing in spring, a nitrogen (N)–phosphorus ($P_2O_5$)–potassium ($K_2O$) compound fertilizer with a ratio of 15:35:10 was applied as the bottom fertilizer, the amount of each community was 3.0 kg, and then no more fertilizer was applied. The surface of the ridgeless plot was raked flat after the ground was turned and fertilized, and the long grass was removed at times during the monitoring period, to reduce surface disturbance as much as possible. At the beginning of the test, soil samples from the soil test cell were collected, the soil structure was analyzed using a laser particle size analyzer (Winner 2308, Jinan micro-nano, Jinan, China), the dry bulk density of the soil was determined via the ring-knife method, and the soil cation exchange amount was determined via the ammonium acetate method [27].

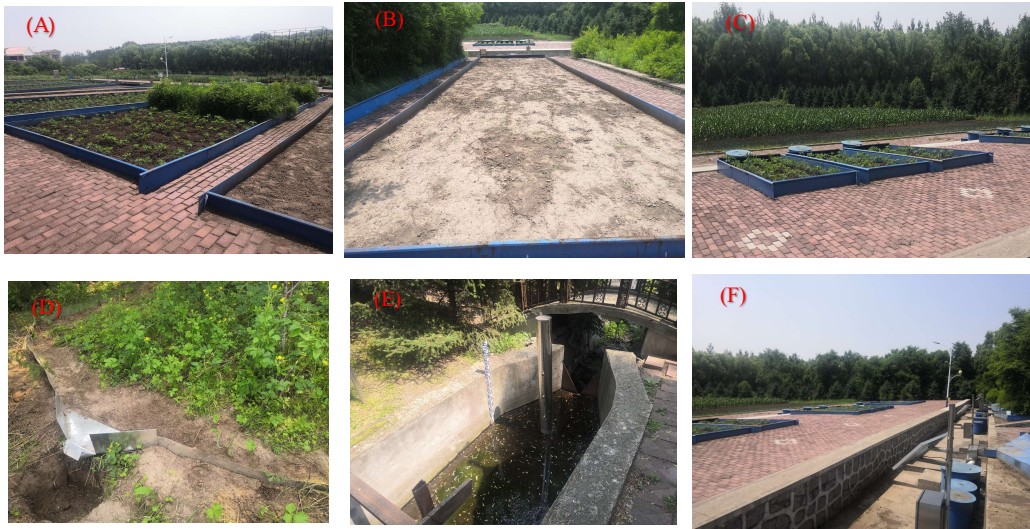

**Figure 2.** The experimental plots and sediment collection device. (**A–C**) represents cross-ridge, no-ridge, and along-the-ridge plots; (**D–F**) represents the precipitation and sediment collection devices.

In addition, the bottom of the runoff community is connected to the rainwater and sediment collection device, to improve the separation efficiency of the soil solution and sediment, and to increase the standing buffer time of the soil solution, the runoff community soil erosion collector adopts a three-stage diversion barrel. The bottom area of the first-stage diversion barrel is 0.5027 $m^2$, the height is 80 cm, and the diversion height of the diversion barrel is 60 cm; the bottom area of the secondary diversion barrel is 0.1964 $m^2$, the height is 60 cm, and the diversion barrel diversion height is 50 cm; the bottom area of the collection barrel is 1967 $cm^2$, and the height is 60 cm. Through the measurement of the total amount of precipitation flow, combined with the area of runoff cells, the depth of the rainfall runoff in the experimental community was obtained [28].

**Table 1.** Basic overview of runoff communities.

| No. | Slope | Ridge Type | Mechanical Composition of the Soil | | | Soil Dry Density $(cm^3 \cdot g^{-1})$ | Initial Moisture Content (%) | Cation Exchange Capability $(cmol \cdot kg^{-1})$ |
| | | | Clay (<0.002 mm) | Silt (0.002–0.02 mm) | Sand (>0.02 mm) | | | |
|---|---|---|---|---|---|---|---|---|
| A1 | 3° | Ridge cultivation along the slope | 27.5 | 42.2 | 31.3 | 1.32 | 25.62 | 8.63 |
| A2 | 3° | Cross-slope-ridge direction | 26.3 | 43.5 | 30.2 | 1.37 | 24.68 | 7.92 |
| A3 | 3° | No-ridge cropping | 28.6 | 44.5 | 26.9 | 1.34 | 26.35 | 7.63 |
| B1 | 5° | Ridge cultivation along the slope | 27.2 | 46.9 | 25.9 | 1.31 | 25.97 | 8.12 |
| B2 | 5° | Cross-slope-ridge direction | 26.9 | 45.8 | 27.3 | 1.29 | 26.79 | 8.35 |
| B3 | 5° | No-ridge cropping | 28.8 | 44.3 | 26.9 | 1.35 | 27.68 | 8.46 |

### 2.3. Rainfall Monitoring Process

During the observation period in 2021, the number of days of rainfall was 69 days, the number of rainfall times was 59, the number of rainfall times leading to the production flow of runoff communities was 17, the number of occasions of erosive rainfall in runoff communities was 21, the erosive rainfall was 430.2 mm, and the erosive rainfall erosion force was 2350.57 MJ·mm/(hm$^2$·h). The maximum daily rainfall is 48.4 mm, the maximum rainfall erosion force is 728.37 MJ·mm/(hm$^2$·h), the annual rainfall erosion force is 2900.86 MJ mm/(hm$^2$·h), and the total rainfall is 600.6 mm. To reveal the mechanism of different rainfall intensities on the flow production, sand production, and nutrient loss in arable land, and to analyze the effect of the blocking and controlling mechanisms of different arable land types on water and fertilizer loss from the perspective of soil hydrodynamics, five typical rainfall processes during the experimental period were selected as research objects. The rainfall characteristics are shown in Table 2.

**Table 2.** Characteristics of typical precipitation processes during the test period.

| Date of Rainfall (y-m-d) | Rainfall (mm) | The Rainfall Lasts for a Long Time (min) | Average Rainfall Intensity $(mm \cdot h^{-1})$ | $I_{30}$ (mm) | Rainfall Erosion $(MJ \cdot mm/(hm^2 \cdot h))$ | Type of Rainfall |
|---|---|---|---|---|---|---|
| 2021-07-02 | 9.8 | 105 | 5.6 | 16.1 | 36.83 | light rain |
| 2021-07-07 | 28.7 | 818 | 2.1 | 31.6 | 221.34 | heavy rain |
| 2021-07-08 | 27.3 | 284 | 5.8 | 28.6 | 202.35 | heavy rain |
| 2021-08-25 | 43.3 | 535 | 4.9 | 39.8 | 445.61 | rainstorm |
| 2021-09-11 | 11.7 | 1459 | 0.5 | 5.8 | 11.05 | moderate rain |

### 2.4. Sample Collection and Analysis

2.4.1. Water Sample and Sediment Separation Method

When collecting the sub-rainfall runoff samples, we first mixed the water samples in the collection tank, and stirred them thoroughly, then quickly collected the water-and-sand mixed samples in washed polyethylene plastic bottles. After the sample had stood for 12 h, the sediment was placed in the oven, and dried at 105 °C for 24 h. The drying sediment was weighed using an electronic balance (with an accuracy of 0.01 g), and the sediment obtained was used in the measurement of sand in the community diameter, and the measurement of nitrogen and phosphorus in the sediment [29,30].

Instantaneous water samples at the outlet of the runoff cell were collected every 0.5 h within the first two hours after rainfall production, and then every 1 h until the end of the runoff. Each sample's sampling volume was 500 mL, and the collected water samples were acidified to pH 1~2, sealed, stored at 4 °C, and then sent to the laboratory for analysis within 24 h, for the determination of the nitrogen and phosphorus content in the runoff [31].



2.4.2. Water Sample and Sediment Analysis Method

The obtained soil runoff water sample was passed through a 0.45 μm filter membrane, and the oxidant prepared using potassium persulfate and sodium hydroxide was added to the filtered water sample, and oxidized and decomposed via heating at 120 °C for 30 min, to ensure that the nitrogenous compounds in the water sample had decomposed into nitrate. Then, the total nitrogen content was measured with an ultraviolet spectrophotometer [32]. In addition, an appropriate amount of the water sample was added to the potassium persulfate solution, and the phosphorus concentration was measured via heating and oxidation at 120 °C for 30 min, to ensure that the phosphorus-containing compounds in the water sample were oxidized and decomposed into orthophosphate, and then counted into an ascorbic acid solution, and thoroughly mixed. Then, the total phosphorus concentration was measured busing a spectrophotometer [33]. In addition, the nitrate nitrogen and ammonium nitrogen content in aqueous solutions were analyzed using flow analyzers (Autoanalyser III, Bran + Luebbe GmbH, Hamburg, Germany), while the available phosphorus was determined directly via the colorimetric colorimetry of antimony ascorbic acid without heat treatment [34]. For the testing and analysis of the total nitrogen, nitrate nitrogen, ammonium nitrogen, total phosphorus, and available phosphorus in the soil, various forms of nitrogen and phosphorus elements need to be extracted from the soil in advance, and then tested and analyzed, according to the above methods.

*2.5. Analysis of the Erosion Mechanism*

The description of the flow regime and properties of slope runoff is usually characterized by hydraulics and erosion sand production hydrodynamic parameters, including the runoff depth, flow velocity, Reynolds number, Froude number, and drag coefficient. Of these, the runoff depth and velocity are the input elements, and are obtained through experimental measurements, while the other parameters are calculated using the relevant open-channel hydraulic equations:

The flow rate measured during the test is only the surface flow rate, and has to be corrected in order to obtain the average flow rate. This is calculated by the following equation:

$$V = kV_{\mathrm{m}} \tag{1}$$

where $V_{\mathrm{m}}$ is the surface flow velocity; $V$ is the mean flow velocity; k is a coefficient of 0.67 for laminar flow, 0.7 for transition flow, and 0.8 for turbulent flow [35].

The Reynolds and Froude numbers are used to determine the flow pattern of the water. The Reynolds number (Re) is used to determine whether the flow is laminar or turbulent. When Re < 500, the flow is laminar; when Re > 500, the flow is turbulent. The Froude number (*Fr*) is the ratio of the inertial force of the flow to the gravitational force, and is used to determine whether the flow is rapid or slow. When *Fr* < 1, the flow is slow; when *Fr* > 1, the flow is fast. The expressions are as follows:

$$\begin{cases} \mathrm{Re} = \frac{VR}{v} \\ Fr = \frac{V}{\sqrt{gh}} \end{cases} \tag{2}$$

where $v$ is the kinematic viscosity coefficient ($\mathrm{cm^2 \cdot s^{-1}}$); $R$ is the hydraulic radius (cm); $V$ is the mean flow velocity ($\mathrm{cm \cdot s^{-1}}$); and $h$ is the runoff depth (m).

The coefficient of resistance (*f*) is a general term for the forces that impede the movement of water from the soil–water interface during the downward movement of runoff, and is expressed as follows:

$$f = \frac{8gRJ}{V^2} \tag{3}$$

where $R$ is the hydraulic radius (m); $J$ is the water surface energy slope ($\mathrm{m \cdot m^{-1}}$); and $V$ is the average flow velocity of the water ($\mathrm{m \cdot s^{-1}}$).

The runoff shear is the main driving force for separating the soil, and dispersing the soil particles and carrying them off the slope. It is calculated as follows:

$$\tau = \gamma R J \tag{4}$$

where $\tau$ is the runoff shear (Pa); and $\gamma$ is the water gravity (N·m$^{-3}$).

There is a significant correlation between water flow power and runoff shear, which is expressed using Equation (5).

$$W = \tau V \tag{5}$$

where $W$ is the flow power (N·m$^{-1}$·s$^{-1}$); and $V$ is the average flow velocity (m·s$^{-1}$).

The concept of the power per unit of flow is based on the conventional sediment transport equation, and defines the power per unit of flow as the product of the flow velocity and the slope drop.

$$\varphi = V J \tag{6}$$

where $\varphi$ is the power per unit of water flow (m·s$^{-1}$); and $J$ is the water surface slope energy (m·m$^{-1}$).

*2.6. Data Processing*

The data processing, plotting, and tabulation used SPSS 22.0 and Sigmaplot 12.5 software. We calculated the mean and standard deviation (SD) for each set of trial data. One-way ANOVA was used to test the differences between treatments at a significance level (*p*) of 0.05 (Duncan's multiple range test).

## 3. Results and Discussion

*3.1. Runoff Sand Effect*

The precipitation and sand effect of the runoff community is shown in Figure 3. Firstly, the analysis of the runoff characteristics under different rainfall intensities showed that, from the first to the fifth rainfall, all three different types of tillage treatment exhibited a gradual increase in runoff depth with the increasing rainfall intensity. This result was positively correlated with the trends in the average rainfall intensity, $I_{30}$, and duration of rainfall in Table 2 above. In addition, in the type A test cell (3°), under the condition of precipitation of 9.8 mm (the first precipitation), the runoff depth of the downslope-ridge test cell was 1.44 mm, while, in the cross-slope-ridge direction and the no-ridge test cell, the runoff depth became 0.11 mm and 3.45 mm, respectively, and the transverse ridge crop effectively inhibited the precipitation flow effect of the sloped cultivated land, while the ridgeless crop increased the runoff depth of the precipitation flow to a certain extent, as in Liu et al. [36]. The proposed contour ridge of the cross-slope ridge can effectively reduce the kinetic energy of rainwater, slow down the erosion and erosion effect of heavy rainfall on slope cultivated land, and reduce the runoff loss on slope cultivated land. When the precipitation increased to 43.3 mm (the fourth precipitation), the soil runoff depth increased by 7.76, 0.69, and 8.50 mm under the three cultivation modes compared with the first precipitation conditions, which also confirmed the findings of Hou et al. [37], that the precipitation intensity affects the slope recharge coefficient of slope cultivated land, and with the increase in precipitation intensity, the soil infiltration rate and cumulative infiltration decrease, and the precipitation production flow increases. At the same time, in the type B test cell (5°), all three different tillage patterns showed further increases in runoff depth relative to the A test cell (3°), which confirmed that, with the increase in the slope cultivated land slope, the runoff depth of the experimental cell showed a significant improvement trend, which, once again, verified the results of Zhao et al. [38]. The stagnation of the slope properties decreases with the increase in the slope, and the soil yield rate and runoff depth show a significant increase trend under the same precipitation intensity.

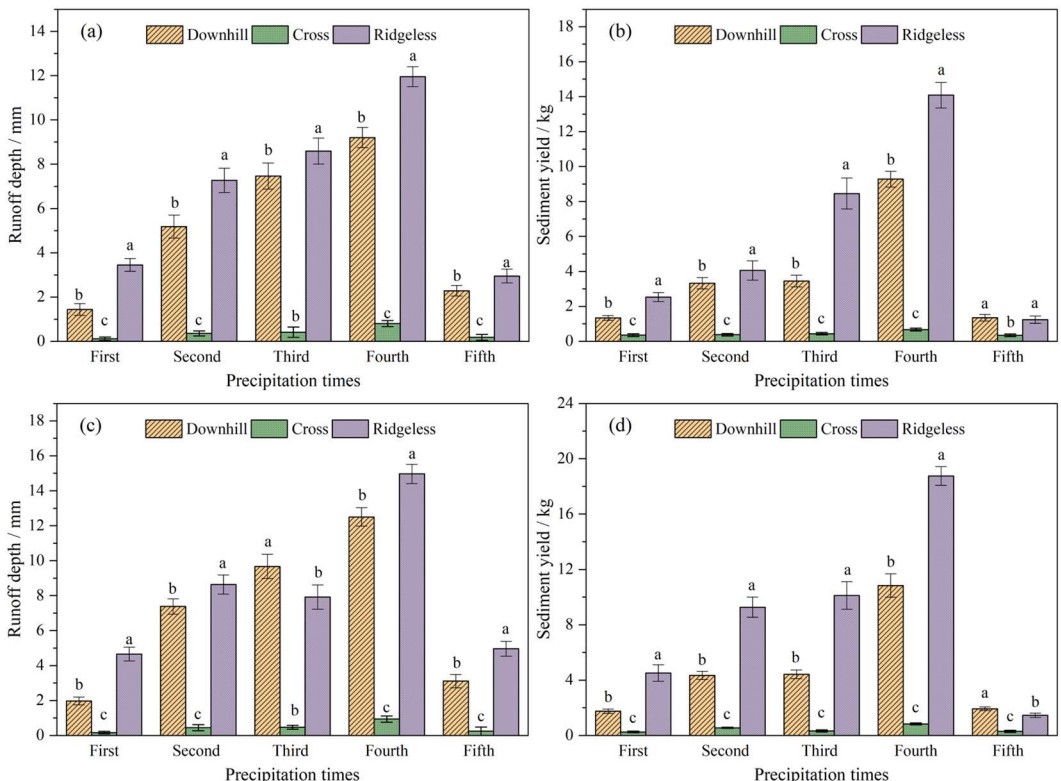

**Figure 3.** Analysis of the soil erosion effect under precipitation conditions. Note: (**a**) the runoff depth at a 3° slope; (**b**) the soil erosion at a 3° slope; (**c**) the depth of runoff at a slope of 5°; and (**d**) the soil loss at a slope of 5°. The different letters indicate significant differences of soil runoff depth and sediment yield ($p < 0.05$).

Farmland soil loss is affected by multiple environmental factors, and atmospheric precipitation production and sediment production have a strong transportation capacity, which has become the main driving force behind topsoil erosion in northern China [39]. Precipitation, topography, and farming patterns are closely related to the soil erosion and loss process [40]. Specific analysis of sediment loss under precipitation erosion shows that, under the conditions of a type A test area (3°) and precipitation of 9.8 mm (the first precipitation), the sediment loss in the downslope-ridge test community is 1.34 kg, while the sediment loss in the cross-slope-ridge direction and the non-ridge test community is 0.35 kg and 2.53 kg, respectively, indicating that the transverse ridge crop reduces the sediment-carrying capacity of the precipitation flow, and effectively inhibits the effect of farmland soil erosion. Luo et al. [41] also proposed in the study that the sediment loss from slope-ridge treatment is greater than that from horizontal slope-ridge treatment during the process of rainfall production flow, and the sediment loss from cross-slope-ridge treatment accounts for only 30~44% of the slope treatment. During the monitoring process, with the increase in the runoff depth, the soil sand production increased, and when the precipitation reached the maximum (the fourth precipitation), the soil and yield increased to 9.27 kg under the slope treatment, while the soil sand yield increased to 0.67 and 14.08 kg under the transverse-ridge and no-ridge treatment, respectively. The long-term field observations of Peng et al. [42] have found that, as the intensity of rainfall increases, the higher the impact capacity of raindrops on topsoil, and the easier it is for soil particles to disperse, meaning that the ability of the water flow to transport sediment is enhanced, thereby increasing the loss of runoff and sediment nutrients.

This section may be divided into subheadings. It should provide a concise and precise description of the experimental results, their interpretation, as well as the experimental conclusions that can be drawn.

### 3.2. Nitrogen and Phosphorus Loss Effects in Runoff Communities

The nitrogen in soil, as an important element replenishment, is conducive to promoting photosynthesis and protein synthesis in crops, accelerating the accumulation of substances and nutrient metabolism in crops, and playing an important role in crop growth [43,44]. With changes in the climate environment, nitrogen is lost in different forms. There are many factors affecting nitrogen loss, and surface runoff is the main driving force behind the migration of dissolved matter in the soil [45]. In this study, the effect of nitrogen loss in the soil erosion caused by rainfall runoff was discussed in depth, and the nitrogen loss in each runoff area showed an upward trend with the increase in precipitation (Figure 4). Comparing the characteristics of soil nitrate nitrogen, ammonium nitrogen, and total nitrogen loss under different rainfall intensity conditions, the soil nitrogen loss gradually increases with rainfall, i.e., the increase in the average rainfall intensity and cumulative rainfall. In addition, the soil nutrient loss also showed significantly different effects on different slopes and tillage patterns. Firstly, in the type A test cell (3°), under the condition of precipitation of 9.8 mm (the first precipitation), the ammonium-nitrogen loss in the downslope-ridge direction test cell was 4.74 g/hm$^2$, while, in the cross-slope-ridge direction and the no-ridge test area, the ammonium nitrogen loss became 1.98 g/hm$^2$ and 10.18 g/hm$^2$, respectively, and the transverse ridge crop effectively inhibited the loss effect of soil nitrogen, while the non-ridge crop increased the nitrogen loss to a certain extent, as reported by [46] The results show that the interception effect of the cross-slope on runoff prolongs the interaction time between runoff and the soil surface, which buys time for the rainwater to fully infiltrate into the soil, and then reduces the erosion, and carries the effect of precipitation runoff on the soil nutrients. When the precipitation increased to 43.3 mm (the fourth precipitation), the soil ammonium nitrogen loss under the three cultivation modes increased by 86.46, 11.46, and 114.30 g/hm$^2$ compared with the first precipitation conditions, indicating that, with the increase in the precipitation intensity, the impact of precipitation runoff on the ground surface was greater, and the enrichment effect of the erosion sediment on the soil nitrogen was increased [47].

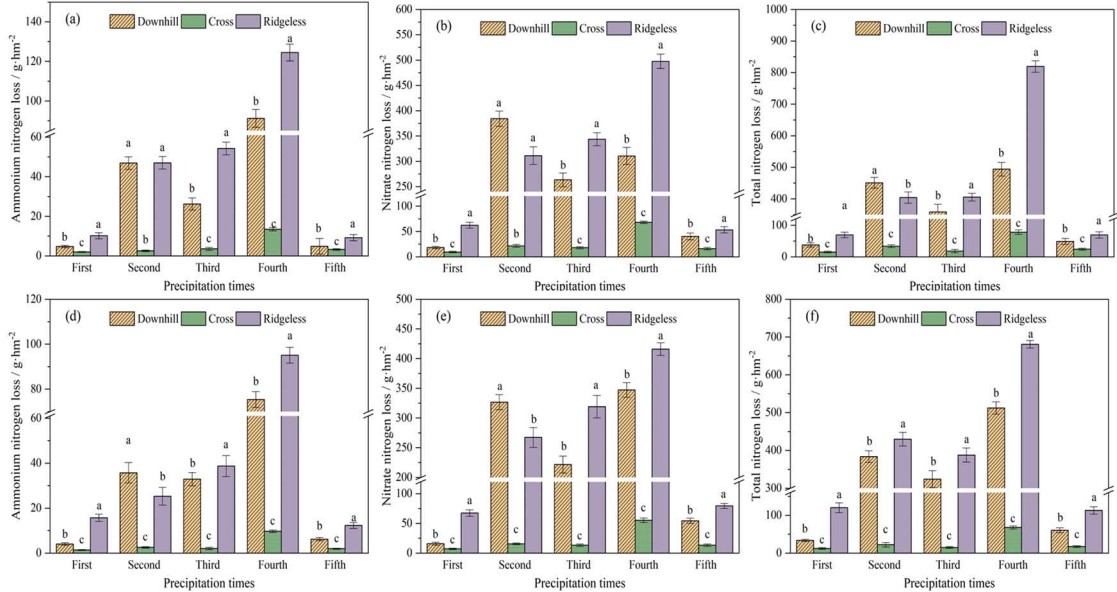

**Figure 4.** Analysis of the soil nitrogen loss effect. Note: (**a**) the soil ammonium nitrogen loss at a 3° slope; (**b**) the soil nitrate nitrogen loss at a slope of 3°; (**c**) the total soil nitrogen loss at a 3° slope; (**d**) the soil ammonium nitrogen loss at a slope of 5°; (**e**) the soil nitrate nitrogen loss at a slope of 5°; and (**f**) the total soil nitrogen loss at a slope of 5°. The different letters indicate significant differences of soil nitrogen content ($p < 0.05$).

On the contrary, in the type B test cell (5°), when the precipitation was 9.8 mm, the ammonium nitrogen loss in the downslope ridge to the test cell became 4.04 g/hm$^2$, which decreased by 14.76% compared with the class A test cell (3°) and, under the conditions of transverse ridge cultivation and no-ridge cropping, the ammonium nitrogen loss decreased by 31.81% and 8.46% relative to the class A test cell (3°), respectively. On the contrary, the loss of nitrogen showed a gradual downward trend. This may be due to the relatively high soil-organic-matter content in the black soil area, the organic complexation between the soil particulate organic functional groups and nitrogen, and a decrease in the free capacity of nitrogen [48]. At the same time, with the increase in the slope, the soil precipitation production rate increases, the runoff time is shortened under the same rainfall intensity, nitrogen and soil colloids cannot be fully desorbed, and the effect of soil nitrogen loss is weakened. This conclusion verifies the research results of Ao et al. [49]; the downslope in different rain intensities is linearly correlated with the amount of soil nitrogen loss per unit area; with the increase in the rain intensity, the soil nitrogen loss increases significantly; and, when the rain intensity is constant, the slope change has little effect on the change rate of nitrogen loss.

The phosphorus in soil plays an important role in plant growth and development, which is related to plant energy biochemical reactions, and is essential for plant cell division and meristem development [50,51]. There are many factors affecting phosphorus loss, such as migration and transformation mechanisms under hydrothermal transport, leaching, volatilization, and surface runoff [52]. The phosphorus loss in runoff communities is shown in Figure 5, and the phosphorus loss in each runoff area shows an upward trend with the increase in precipitation. Firstly, during the five regular rainfall events, the loss of soil-adequate phosphorus and total phosphorus gradually increased with the increased rainfall intensity. Taking the para-tillage treatment as an example, when the rainfall intensity increased from 9.8 to 43.3 mm, the changes in the adequate soil phosphorus under the slope conditions of 3° and 5° were 4.28~68.87 g/hm$^2$ and 5.53~102.78 g/hm$^2$. In the type A, we found that the transverse ridge crop effectively inhibited the loss effect of soil phosphorus, while the ridgeless crop increased the phosphorus loss to a certain extent, just as Bayad et al. [53] reported. The study confirmed that the coupled straw mulching of cross-slope-ridge farming can effectively reduce the total phosphorus loss of surface runoff from slope cultivated land by 36.84~79.66% compared with conventional tillage treatment and single optimized tillage treatment. When the precipitation increased to 43.3 mm (the fourth precipitation), the soil particulate phosphorus loss under the three cultivation modes increased by 68.87, 8.07, and 99.84 g/hm$^2$, respectively, compared with the first precipitation conditions, which may be attributed to the fact that phosphorus is mainly present in the soil surface layer in the granular state, which is easily splashed by raindrops during rainfall, and accompanied by runoff erosion loss. At the same time, in the type B test cell (5°), when the precipitation was 9.8 mm, the loss of granular phosphorus in the downslope ridge to the test cell became 5.53 g/hm$^2$, 1.63, and 11.41, increased by 29.20% compared with the class A test cell (3°), and under the conditions of transverse ridge cultivation and no ridge cropping, the loss of granular phosphorus increased by 45.53% and 53.98% compared with the class A test cell (3°), respectively, with the increase in the slope cultivated land slope. The loss of particulate phosphorus in the experimental community showed a significant growth trend.

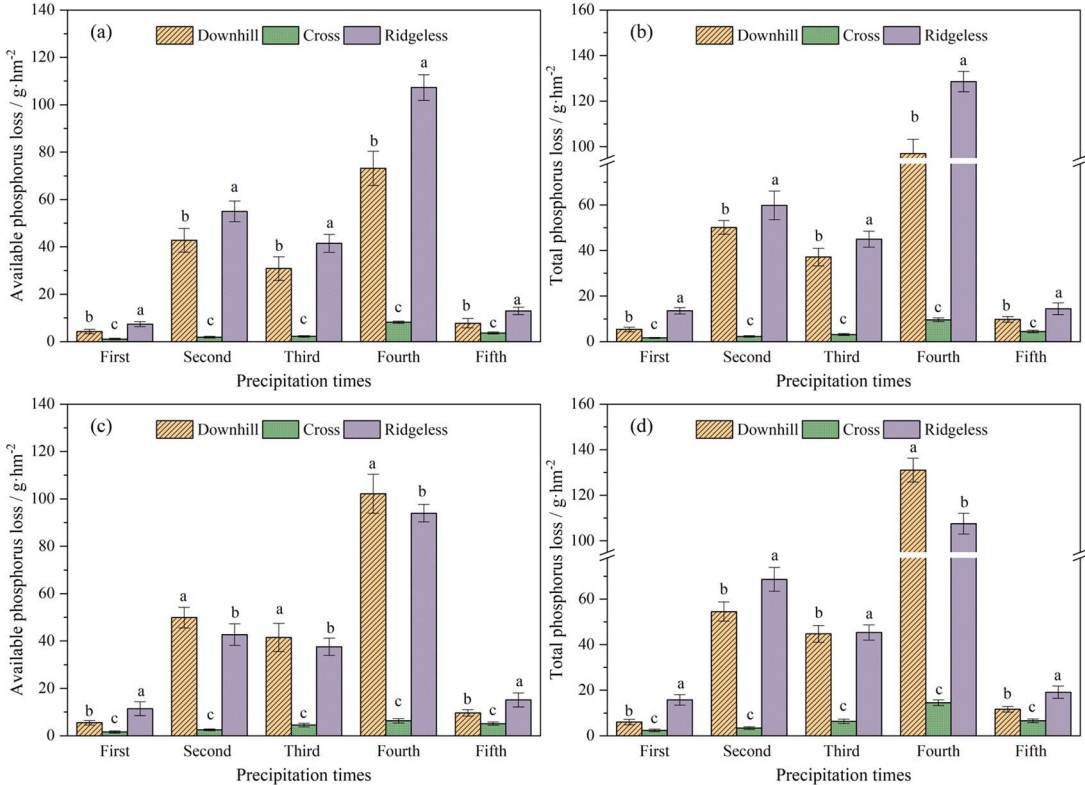

**Figure 5.** Analysis of the soil phosphorus loss effect. Note: (**a**) the soil particulate phosphorus loss at a slope of 3°; (**b**) the total soil phosphorus loss at a 3° slope; (**c**) the soil particulate phosphorus loss at a slope of 5°; and (**d**) the total soil phosphorus loss at a 5° slope. The different letters indicate significant differences of soil phosphorus content ($p < 0.05$).

### 3.3. Analysis of Influencing Factors behind Water and Fertilizer Loss in Runoff Communities

To further reveal the synergistic effect of soil water and fertilizer loss in sloped cultivated land, the response relationship between cultivated land runoff characteristics and nitrogen and phosphorus loss processes was further explored, as shown in Figure 6.

The loss of soil nitrogen and phosphorus increased with the increase in runoff depth and soil erosion, showing a linear correlation (the significance passed the $p < 0.05$ test), and the slope of the fitting curve reflected the carrying capacity of soil erosion to soil nutrients. Firstly, the slope of the fitting curve between the soil runoff depth and ammonium nitrogen loss is 6.08, while the slope of the fitting curve between the runoff depth and soil nitrate nitrogen loss is 32.98, which shows a significant improvement trend compared with ammonium nitrogen, indicating that the transport capacity of soil runoff for nitrate nitrogen is stronger than that for ammonium nitrogen. This also verifies that the ammonium ions proposed by [54] are easily adsorbed by negatively charged soil particles when they move with the runoff, while the adsorption performance of nitrate ions and soil particles is weakened, and the synergistic effect on rainfall washing is strong. In addition, with the increase in precipitation intensity, raindrop splashing and runoff continue to destroy large-grained aggregates in the soil, forcing the release and decomposition of ammonium ions in soil aggregates, which aggravates the loss effect of ammonium ions in farmland soil [55]. Analyzing the correlation between the soil runoff depth and phosphorus loss, it can be seen that, with the increase in the soil runoff depth, the soil available phosphorus and total phosphorus loss show a clear increasing trend, indicating that the response effect of soil phosphorus migration on the soil runoff gradually increases [56].

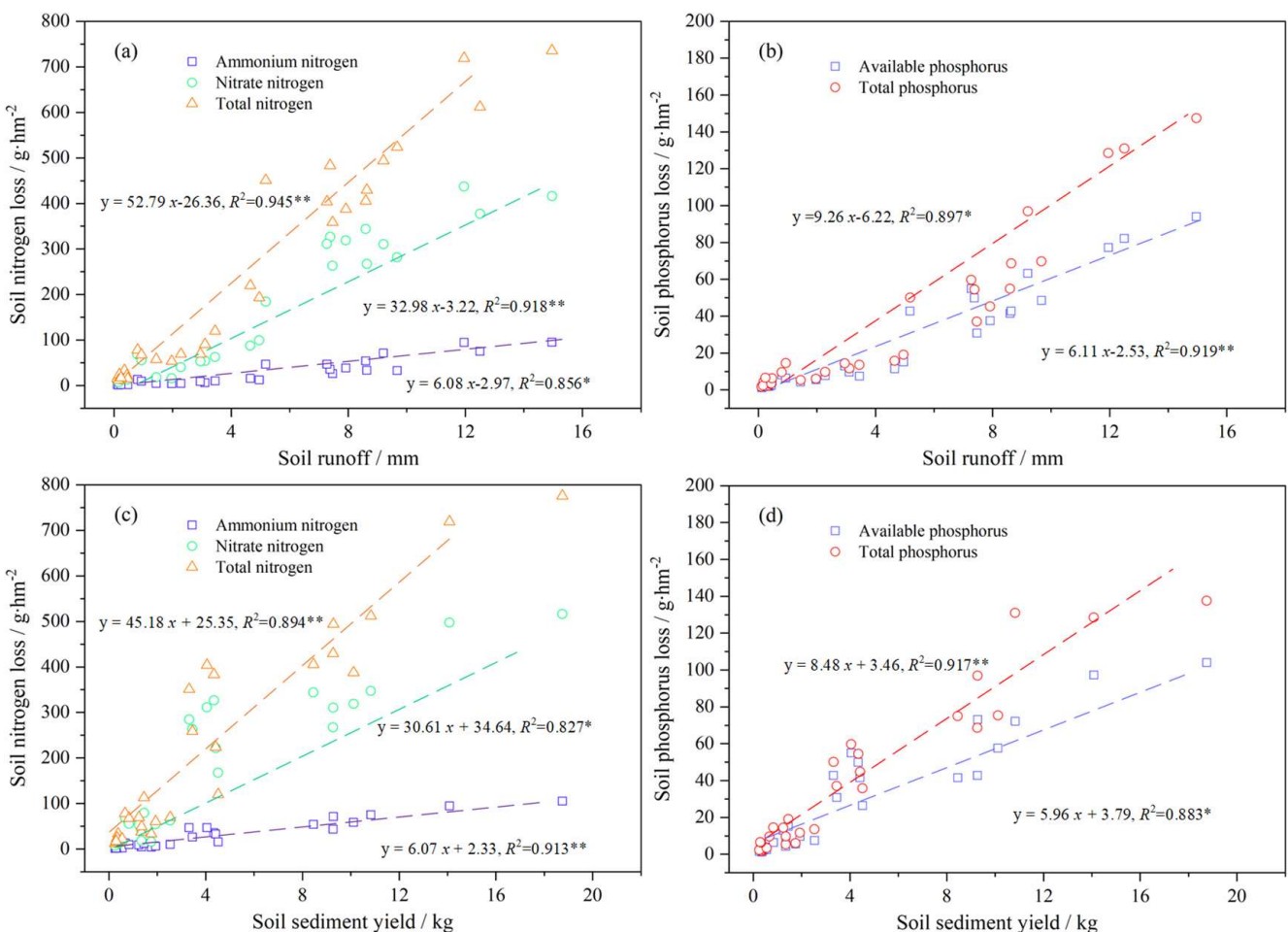

**Figure 6.** The synergistic effect between soil erosion and soil fertility diffusion. Note: (**a**) the relationship curve between soil nitrogen loss and runoff; (**b**) the relationship curve between soil phosphorus loss and runoff; (**c**) the relationship curve between soil nitrogen loss and soil loss; and (**d**) the relationship curve between soil phosphorus loss and soil loss. "**\***" represents that the fitting degree passes the significance test of $p < 0.01$; "*\**" represents that the fitting degree passes the significance test of $p < 0.05$.

The slope of the fitting curve between soil erosion and soil nitrogen and phosphorus loss reflects the nutrient content carried by soil erosion; firstly, the slope of the fitting curve between soil erosion and ammonium nitrogen loss is 6.07, and similarly, the transport capacity of nitrate nitrogen and total nitrogen is significantly improved, and the slope of the fitting curve between soil erosion and nitrate nitrogen and the total nitrogen loss becomes 30.61 and 45.18, respectively, which further verifies that soil loss has a strong carrying effect on nitrate nitrogen. Similarly, the increase in soil erosion also increased the phosphorus loss effect, indicating that there is a strong coupling synergistic effect between phosphorus migration pathways and soil particles, and that soil erosion will lead to obvious fertility loss consequences [57,58].

In addition, the significance analysis of the influence effect of the precipitation, ridge direction, slope, and other factors on soil water and fertilizer loss was further carried out, and the results are shown in Table 3. Firstly, the analysis of single factors had a significant effect on the depth of runoff, sediment yield, and nutrient loss in hilly cultivated land ($p < 0.001$), which fully confirmed the above statistical analysis results, confirming that the rainfall factor is the main driving factor behind the occurrence and development of runoff erosion, and the cultivated land slope and ridge direction affect the hydraulic characteristics of surface runoff [59]. Especially in slope farming, a large catchment area will be formed in the rainy season, and the erosion force caused by surface runoff is strong,

and the cultivated land very easily forms fine ditches and shallow ditches on the slope surface, resulting in serious soil erosion and the thinning of the black soil layer, year by year [60]. When the two factors interact, their influence on the soil water and fertilizer loss process is weakened, especially the loss of ammonium nitrogen, and the combined effect of the precipitation and slope has no significant effect on it. When the three factors interacted, the rainfall erosion only had a significant effect on the loss of soil nitrate nitrogen and available phosphorus ($p < 0.05$), but had a weak effect on other indicators. This also shows that soil water and fertilizer loss under rainfall conditions is a very complex process, which is affected by many factors and conditions [61].

**Table 3.** Analysis of the influencing factors on soil water and fertilizer loss.

| Treatment | Runoff Is Deep | | Sand Production | | Ammonium Nitrogen | | Nitrate | | Total Nitrogen | | Available Phosphorus | | Total Phosphorus | |
|---|---|---|---|---|---|---|---|---|---|---|---|---|---|---|
| | F-Value | Sig. | F-Value | Sig. | F-Value | Sig. | F-Value | Sig. | F-Value | Sig. | F-Value | Sig. | F-Value | Sig. |
| Rainfall | 178.53 | 0.000 ** | 257.23 | 0.000 ** | 287.67 | 0.000 ** | 351.78 | 0.000 ** | 457.67 | 0.000 ** | 231.89 | 0.000 ** | 387.96 | 0.000 ** |
| Ridge direction | 152.36 | 0.000 ** | 221.56 | 0.000 ** | 89.36 | 0.000 ** | 125.56 | 0.000 ** | 172.35 | 0.000 ** | 112.45 | 0.000 ** | 56.35 | 0.000 ** |
| slope | 85.65 | 0.037 * | 159.36 | 0.025 * | 112.32 | 0.072 | 135.64 | 0.114 | 86.36 | 0.089 | 72.36 | 0.023 * | 22.35 | 0.017 * |
| Rainfall × ridge direction | 56.48 | 0.005 ** | 37.58 | 0.002 ** | 54.16 | 0.000 ** | 52.12 | 0.000 ** | 35.64 | 0.000 ** | 28.56 | 0.001 ** | 35.48 | 0.000 ** |
| Ridge direction × slope | 65.69 | 0.008 ** | 52.13 | 0.025 * | 32.11 | 0.015 * | 21.68 | 0.015 * | 63.56 | 0.011 * | 42.15 | 0.007 ** | 21.48 | 0.005 ** |
| Rainfall × slope | 19.74 | 0.033 * | 12.48 | 0.013 * | 12.45 | 0.079 | 9.36 | 0.046 * | 14.58 | 0.028 * | 16.35 | 0.034 * | 14.65 | 0.027 * |
| Rainfall × ridge direction × slope | 15.36 | 0.065 | 6.35 | 0.079 | 3.63 | 0.081 | 4.56 | 0.022 * | 5.28 | 0.137 | 5.27 | 0.037 * | 3.26 | 0.143 |

Note(s): "**" represents that the correlation passes the significance test of $p < 0.01$; "*" represents that the correlation passes the significance test of $p < 0.05$.

## 4. Discussion

### 4.1. Analysis of Rainfall-Runoff Processes

In order to further analyze the influence of precipitation processes on the runoff generation and erosion and sand production processes, the fourth rainfall was used as an example, to explore the trend in rainfall runoff with rainfall ephemeris under different tillage patterns, and the measured runoff reduction rate is shown in Figure 7. During the initial rainfall period, the soil water content was unsaturated, and rainwater infiltrated rapidly, so the runoff reduction rate was at a high level. As time passes, the soil gradually reaches saturation, and the soil rainfall runoff gradually tends to be stable, and at a lower level, and the erosion effect of runoff on the sediment increases. This also verifies the findings of the empirical study of Zhang et al. [62], who reported that soil runoff and sand production show linear and S-curve growth trends, respectively, with increasing rainfall ephemeris. In addition, through a comparison of the trends in the soil runoff reduction rate under different tillage patterns, it can be seen that, during the initial period, the rate of reduction of the rainfall runoff was greatest in the cross-monopoly tillage treatment, followed by the smooth-monopoly tillage, while the lowest reduction rate was found in the no-monopoly tillage treatment and, at the end of precipitation, the rate of reduction of the soil rainfall runoff was, in descending order, cross-monopoly tillage > smooth-monopoly tillage > no-monopoly tillage. When St. Gerontidis et al. [63] studied the effect of down-slope tillage versus contour tillage on soil-particle displacement, it was found that contour tillage effectively inhibited the soil water and fertilizer loss process. In addition, the rate of reduction of soil rainfall runoff decreased as the slope of the cultivated land increased, indicating a gradual increase in the runoff and sand production effects of the precipitation processes.

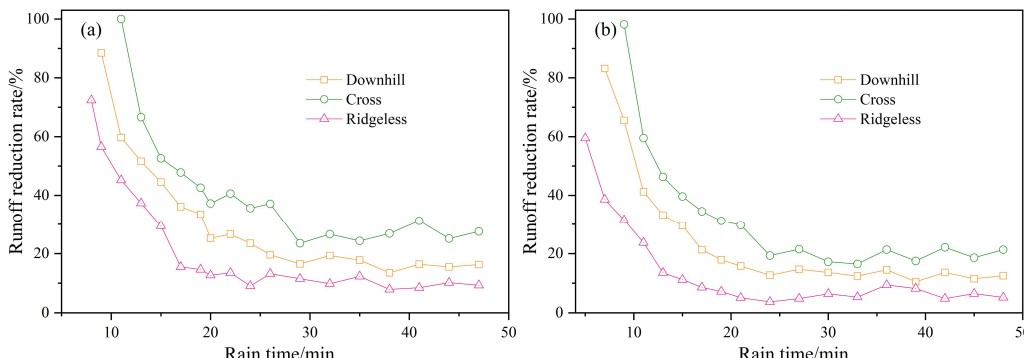

**Figure 7.** Characteristics of runoff reduction rates with the rainfall duration. Note: (**a**) the soil runoff reduction at a slope of 3°; (**b**) the soil runoff reduction at a slope of 5°.

### 4.2. Hydrodynamic Characteristics of Rainfall Runoff

The characteristics of the changes in the hydraulic parameters of rainfall runoff from sloping cultivated land are shown in Table 4. For the runoff Reynolds coefficient (Re), the soil runoff Reynolds coefficient was 96.52 when the slope was 3° and the precipitation was 9.8 mm, and it increased gradually with the increase in precipitation. Under cross-slope tillage, the soil Reynolds coefficients show a significant decrease compared to the down-slope treatment, which confirms that cross-slope tillage has a greater effect on the generation of rainfall runoff. In addition, under the no-monopoly treatment, the soil runoff Reynolds coefficient increased significantly, with the overall level fluctuating between 153.41 and 517.46, and the runoff state changed from laminar flow to turbulent flow. The higher the Reynolds coefficient, the more turbulent the runoff, the greater the erosive transport capacity of the runoff, and the more easily the particles on the slope are displaced [64]. The trend characteristics of the Reynolds coefficients again confirm that para-tillage and cross-tillage alter the flow patterns of rainfall runoff, and that cross-tillage is the most effective in reducing the effect of the soil-flow rates [65]. Similarly, analysis of the Four-drinier number shows that, as the amount of rainfall increases, the soil runoff changes from slow to fast, and cross-monopoly tillage is most effective in reducing the Fourdrinier number by 56.14%, to 64.38%, compared to no-monopoly tillage. On the contrary, with increasing rainfall, the soil runoff resistance coefficient showed a gradual decrease, mainly because the larger the Reynolds coefficient, the stronger the turbulence of the rainfall runoff, and the greater the erosive transport capacity of the runoff, meaning the slope particles migrate more easily and, therefore, the resistance to runoff along the course is extremely reduced [66].

**Table 4.** Characteristics of the hydrodynamic parameters.

| Slope | Rainfall (mm) | Re | | | Fr | | | f | | |
|---|---|---|---|---|---|---|---|---|---|---|
| | | Downhill | Cross | Ridgeless | Downhill | Cross | Ridgeless | Downhill | Cross | Ridgeless |
| 3° | 9.8 | 96.52 | 23.78 | 153.41 | 0.42 | 0.16 | 0.39 | 43.62 | 86.33 | 24.56 |
| | 28.7 | 239.94 | 51.35 | 382.93 | 0.49 | 0.25 | 0.57 | 35.24 | 65.27 | 18.61 |
| | 27.3 | 218.86 | 45.27 | 352.24 | 0.54 | 0.23 | 0.62 | 32.45 | 59.34 | 15.57 |
| | 43.3 | 276.55 | 60.64 | 517.46 | 0.63 | 0.26 | 0.73 | 28.36 | 51.45 | 13.91 |
| | 11.7 | 110.47 | 26.42 | 180.41 | 0.39 | 0.18 | 0.47 | 38.63 | 76.86 | 21.98 |
| 5° | 9.8 | 104.92 | 27.41 | 175.03 | 0.39 | 0.19 | 0.45 | 39.41 | 81.52 | 21.32 |
| | 28.7 | 263.41 | 34.85 | 434.07 | 0.64 | 0.34 | 0.69 | 27.46 | 60.58 | 15.67 |
| | 27.3 | 229.61 | 47.61 | 454.70 | 0.68 | 0.31 | 0.75 | 29.36 | 55.23 | 14.75 |
| | 43.3 | 389.53 | 69.41 | 567.41 | 0.76 | 0.39 | 0.82 | 24.16 | 48.26 | 11.82 |
| | 11.7 | 135.38 | 32.04 | 208.24 | 0.45 | 0.21 | 0.62 | 35.21 | 73.64 | 19.71 |

### 4.3. Rainfall Erosion Force Effect

Again using the fourth rainfall as an example, the soil runoff shear, water-flow power, and unit water-flow power for different tillage patterns are shown in Figure 8. As for the runoff shear: at a slope of 3°, the soil runoff shear was 0.72 Pa and 0.88 Pa for the down-slope and no-monopoly tillage patterns respectively, while the cross-monopoly pattern showed a significant decrease. At the same time, the soil runoff shear increased as the slope of the tillage field increased. This, again, explains the synergistic effect of soil runoff and sediment loss in the para-tillage and no-monopoly cropping patterns, with more severe soil water and fertilizer loss, while the cross-monopoly cropping pattern effectively suppresses rainfall-runoff shear, thereby reducing the sediment-carrying capacity of the soil [67,68]. In addition, the soil runoff water power was significantly increased in the smooth and no-monopoly treatments compared to the cross-monopoly treatment, suggesting that as the water shear increases, the bond between soil particles decreases, the soil shear strength decreases, and the dispersed soil particles are more likely to migrate with the runoff, increasing the runoff water power [69].

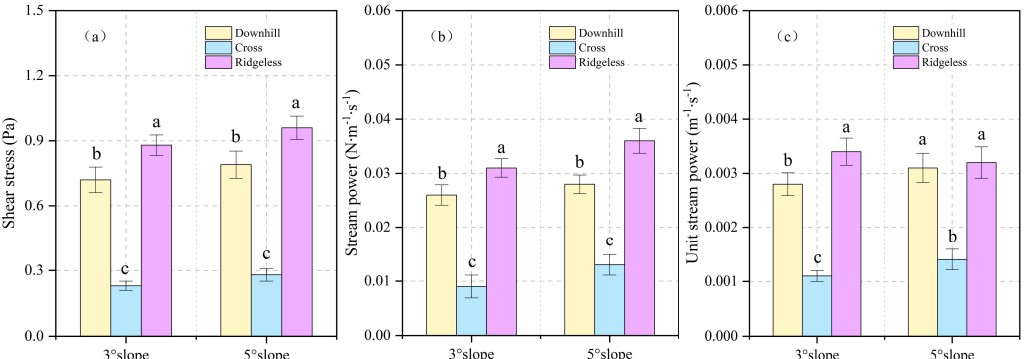

**Figure 8.** Characterization of the rainfall erosion forces. Note: (**a**) runoff shear; (**b**) flow power; and (**c**) unit flow power. The different letters indicate significant differences of rainfall erosion forces ($p < 0.05$).

## 5. Conclusions

The slope, ridge direction, and precipitation intensity of cultivated land under rainfall conditions in the northeast black soil area affect the soil production and flow process to varying degrees, and the cross-slope-ridge direction effectively hinders the migration path of the water flow, changes the rainwater distribution mode, and inhibits the ineffective loss of water. However, with the increase in the slope and precipitation intensity, the depth and runoff rate of the soil runoff increased significantly ($p < 0.05$), and the hydraulic erosion effect of precipitation production on the soil increased correspondingly. The impact of rainwater carries a large amount of sediment, which triggers the effect of soil nutrient loss. The ridge direction of the cross-slope can not only reduce the slope runoff, but also reduce the washing of the soil surface and the sediment-carrying capacity, increase water infiltration in the soil, improve the soil structure, and reduce soil erosion. It is particularly noteworthy that the increase in rainfall intensity will enhance the loss of soil nitrogen and phosphorus under various tillage modes; however, with the increase in the soil slope, the carrying capacity of rainfall-runoff on soil nitrogen has decreased, which confirms that the increase in the rainfall production flow rate has no significant effect on the erosion process of soil nitrogen. The comprehensive analysis results show that the individual regulation of the cultivated land slope, ridge direction, and precipitation intensity has a significant impact on the soil water and fertilizer loss, and the degree of influence is closely related to the soil itself and the regulation mode, while the mechanism of the multi-factor combination on the soil erosion effect is weakened.

Although the one-way analysis of variance method can be used to effectively analyze the migration pattern of soil nitrogen and phosphorus in hilly diffuse arable land during rainfall runoff, there is also room for improvement, going forward For example, in the farm-

land nonpoint source pollution diffusion path identification problem, farmland water and soil environment health control, which requires accurate diagnosis and identification technology to improve the accuracy of the description of the pollution-migration process. In addition, the complexity of hydrological process in cold regions also increases the uncertainty of pollutant diffusion. Therefore, there is a need for more effective identification, prediction, and management technology, such as artificial intelligence theory, mechanism model simulation, and a policy support system, which can help to effectively prevent the pollution effects of human agricultural activities on the soil and water environment, and allow the cultivated land of the northeastern black soil slopes to be healthy and sustainable.

**Author Contributions:** Conceptualization, T.L.; methodology, F.M.; software, J.W.; validation, J.W.; formal analysis, P.Q.; investigation, N.Z.; resources, W.G.; data curation, J.X.; writing—original draft preparation, T.L.; writing—review and editing, T.D.; funding acquisition, F.M. All authors have read and agreed to the published version of the manuscript.

**Funding:** This research was supported by the 2021 Open Fund of the State Key Laboratory of Urban Water Resources and Environment, Harbin Institute of Technology (ES202116). This research was financially supported by Heilongjiang Touyan Team.

**Data Availability Statement:** Not applicable.

**Conflicts of Interest:** The authors declare no conflict of interest.

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
