# Peer review of "Study on the Mechanism of Rainfall-Runoff Induced Nitrogen and Phosphorus Loss in Hilly Slopes of Black Soil Area, China"

_water, doi:10.3390/w15173148_

Round 1

Reviewer 1 Report

Major review:

1 The introduction part is poorly organized. Despite the title is mechanism of rainfall runoff-induced Nitrogen and Phosphorus loss, how the rainfall-runoff impacted the Nitrogen and Phosphorus loss is barely mentioned. Erosion, is just the result of rainfall-runoff.

2 This paper used too many long sentences with confusing words, made it very hard to understand. Both the logic and flow is poor.

3 Only two different slopes of 3 and 5 were set in this paper. The composition of the soil is not supposed to generate a significant runoff yield on runoff. I suggest increase different slopes.

4 Only 5 different rainfall events were monitored. And I dont think these rainfall event were enough to reveal the mechanism of rainfall runoff-induced Nitrogen and Phosphorus loss and to support the conclusions.

5 In my opinion, this paper focused on the observation of the experimental results, instead of the  the mechanism of rainfall runoff-induced Nitrogen and Phosphorus loss. More discussion and conclusion should focus on this instead of just description of the observation results.

Minor review:

1 Many parts of the paper are very confusing and lack of logic.

2 Line 11: China can generate larges amount of (such as Nitrogen and Phosphorus), This sentence is not complete.

3 Line 51: The word however here is not appropriate.

4 Line 51 and 52: There were two words of irrational and irrigative.

5 Line 54 to 56 is very confusing. The use of long sentence made the whole paper hard to read. It should be  lead to.. instead of  lead“

6 Line 85 to 86: Very confusing: eroded sediment between fine trenches is continuously transported to fine trenches

7 Line 150-153: This sentence is too long. Reorganize.

8 Line 229: I never heard of the term called”runoff hydraulics”

9 In section 5: Using . instead of 。 This part should include the reference and should be simplified since it is kind of a common sense.

10 In runoff-sand effect: Line 276-277: Precipitation and sand effect of runoff community, as shown in Figure 2, the runoff depth of each runoff community increased with the increase of precipitation. We know this result even we dont do any experiments. Please clarify your findings different from other studies.

Major review:

1 The introduction part is poorly organized. Despite the title is mechanism of rainfall runoff-induced Nitrogen and Phosphorus loss, how the rainfall-runoff impacted the Nitrogen and Phosphorus loss is barely mentioned. Erosion, is just the result of rainfall-runoff.

2 This paper used too many long sentences with confusing words, made it very hard to understand. Both the logic and flow is poor.

3 Only two different slopes of 3 and 5 were set in this paper. The composition of the soil is not supposed to generate a significant runoff yield on runoff. I suggest increase different slopes.

4 Only 5 different rainfall events were monitored. And I dont think these rainfall event were enough to reveal the mechanism of rainfall runoff-induced Nitrogen and Phosphorus loss and to support the conclusions.

5 In my opinion, this paper focused on the observation of the experimental results, instead of the  the mechanism of rainfall runoff-induced Nitrogen and Phosphorus loss. More discussion and conclusion should focus on this instead of just description of the observation results.

Minor review:

1 Many parts of the paper are very confusing and lack of logic.

2 Line 11: China can generate larges amount of (such as Nitrogen and Phosphorus), This sentence is not complete.

3 Line 51: The word however here is not appropriate.

4 Line 51 and 52: There were two words of irrational and irrigative.

5 Line 54 to 56 is very confusing. The use of long sentence made the whole paper hard to read. It should be  lead to.. instead of  lead“

6 Line 85 to 86: Very confusing: eroded sediment between fine trenches is continuously transported to fine trenches

7 Line 150-153: This sentence is too long. Reorganize.

8 Line 229: I never heard of the term called”runoff hydraulics”

9 In section 5: Using . instead of 。 This part should include the reference and should be simplified since it is kind of a common sense.

10 In runoff-sand effect: Line 276-277: Precipitation and sand effect of runoff community, as shown in Figure 2, the runoff depth of each runoff community increased with the increase of precipitation. We know this result even we dont do any experiments. Please clarify your findings different from other studies.

Reviewer 2 Report

The essay is very timely. Well edited, references are correct. 

Tables and figures are schematic and interpretable.

Accept in present form.

Author Response

On behalf of all the authors, thank you very much for your review of our manuscript. You have provided many constructive comments and improved the quality of the manuscript. 

Reviewer 3 Report

the article entitled "Study on the mechanism of rainfall runoff-induced Nitrogen and Phosphorus loss in hilly slopes of black soil area, China" presents an approach on erosion processes in mountainous regions that is extremely important and has potential for application in other regions of the world, with crops in similar reliefs. The present version presents itself well contextualized regarding the literature review.

Some suggested changes:

- Figure 1 should show the location of the experimental area/region in China;

- Present details and images of the rainwater and sediment collection device;

- Insert a new Figure with images of the experimental plots (runoff communities) and their respective cultivation conditions;

- Standardize the nomenclature of sand, silt and clay (follow international standards for soil science) - terms used in Table 1 are unusual.

- Figures 2, 3 and 4 should be changed in terms of the X axis. Use variables from Table 2 to improve the analyzes (average rainfall intensity, I30, duration of rainfall - this will be better than the order of analysis of events) ;

- Provide corrections regarding citations and references, as per MDPI instructions.

After these minor adjustments, the paper can be accepted for publication.

Reviewer 4 Report

The paper „Study on the mechanism of rainfall runoff-induced Nitrogen and Phosphorus loss in hilly slopes of black soil area, China” is current and very well structurate.

The authors highlighted very well the migration pattern of soil nitrogen and phosphorus in hilly and diffuse arable land in the northeastern black soil region during rainfall runoff.

The perspectives in the continuation of research and trials are also presented very well.

I propose to publish the paper in present form.

Author Response

(The authors gave the same response as above.)

Reviewer 5 Report

The manuscript entitled "Study on the mechanism of rainfall runoff-induced Nitrogen and Phosphorus loss in hilly slopes of black soil area, China" presents a study that highlights the relationship between rainfall erosion and soil water and fertilizer loss. Overall, it's well-written and enriched, except for the abstract, which I find a little long. As well, I'd like the authors to include field photos if possible.

Round 2

Reviewer 1 Report

Even the authors tried to explain the questions, they actually didn't try to add the necessary  experiments. The data in this paper is not sufficient. 

Even the authors tried to explain the questions, they actually didn't try to add the necessary  experiments. The data in this paper is not sufficient. 
